# Adipocyte-Derived Extracellular Vesicles: State of the Art

**DOI:** 10.3390/ijms22041788

**Published:** 2021-02-11

**Authors:** Sophie Rome, Alexia Blandin, Soazig Le Lay

**Affiliations:** 1CarMeN Laboratory, INSERM/1060- INRAE/1397, University of Lyon, Lyon-Sud Faculty of Medicine, 69310 Pierre Benite, France; 2Institute of Functional Genomic of Lyon (IGFL), ENS, CNRS UMR 5242, University of Lyon, 69364 Lyon, France; 3Université de Nantes, CNRS, INSERM, L’Institut du Thorax, F-44000 Nantes, France; alexia.blandin@inserm.fr; 4Univ Angers, SFR ICAT, F-49000 Angers, France

**Keywords:** adipocytes, extracellular vesicles, exosomes, obesity, diabetes, therapy

## Abstract

White adipose tissue (WAT) is involved in long-term energy storage and represents 10–15% of total body weight in healthy humans. WAT secretes many peptides (adipokines), hormones and steroids involved in its homeostatic role, especially in carbohydrate–lipid metabolism regulation. Recently, adipocyte-derived extracellular vesicles (AdEVs) have been highlighted as important actors of intercellular communication that participate in metabolic responses to control energy flux and immune response. In this review, we focus on the role of AdEVs in the cross-talks between the different cellular types composing WAT with regard to their contribution to WAT homeostasis and metabolic complications development. We also discuss the AdEV cargoes (proteins, lipids, RNAs) which may explain AdEV’s biological effects and demonstrate that, in terms of proteins, AdEV has a very specific signature. Finally, we list and suggest potential therapeutic strategies to modulate AdEV release and composition in order to reduce their deleterious effects during the development of metabolic complications associated with obesity.

## 1. Introduction

White adipose tissue (WAT) distributes in discrete anatomical depots identified as subcutaneous adipose tissue (SAT) or visceral adipose tissue (VAT). The expansion of both depots contributes to obesity. Nonetheless, the development of metabolic complications is preferentially associated with VAT expansion [1]. WAT represents overall 10–15% of total body weight in a healthy human and constitutes the main energy supply in the body, being mobilized according to the body’s needs (for review [2]). WAT also has endocrine functions and secretes many peptides, hormones and steroids that participate in its homeostatic role, especially in carbohydrate–lipid metabolism regulation. Communication with other key metabolic tissues is moreover achieved through dense vascularization and innervation, all organized in a metabolically active connective tissue [2,3]. High plasticity of WAT to adapt and expand in response to energy surplus involved increased adipocyte size (hypertrophia) and/or recruitment and proliferation of precursor cells (hyperplasia) in combination with vascular and extracellular matrix remodeling.

At the cellular level, the storage of energy under the form of lipids, namely triacylglycerols, is ensured by adipocytes in a huge lipid droplet filling the cytoplasm. Fat cells are then highly expandable, with a size that can reach up to 150 µm in obese patients [4]. The rapid expansion of WAT in response to nutrient overload is signed by a profound remodeling of fat, involving all cellular components of this organ. Many cellular stresses associated with excessive fat mass development (local hypoxia, inflammation and oxidative or endoplasmic reticulum stresses) have indeed been associated with an increased macrophage infiltration within WAT, representing up to 40% of the total cell content in case of massive obesity [5]. This participates in the chronic inflammation state associated with obesity, which, in addition to impacting WAT remodeling, also promotes the development of insulin resistance, a major metabolic dysfunction associated with obesity [2,3,6].

In this context, adipocyte-derived extracellular vesicles (AdEVs) have recently been highlighted as important actors of intercellular communication that participate in metabolic responses to control energy flux and immune response. In this review, we will focus on the role of AdEV in the cross-talks between the different cellular types composing WAT with regard to their contribution to WAT homeostasis and metabolic complications development. We will also discuss the AdEV cargoes (RNA, lipids, proteins) that participate in AdEV’s biological effects on recipient cells. Finally, we will describe potential therapeutic strategies to modulate AdEV release and composition in order to reduce their deleterious effects during the development of metabolic complications associated with obesity.

## 2. Generalities on Extracellular Vesicles

Extracellular vesicles (EVs) have long been viewed as conveyors of cellular waste used by the cell to get rid of harmful or unnecessary molecules [7]. EVs designate nanovesicles derived from cells or organelle membranes that are secreted into the extracellular medium and circulate in all body fluids (blood, lymph, urine, milk, saliva, tears, etc.) [8,9]. EVs are now recognized as vectors of biological material (proteins, lipids and nucleic acids) and are able to target and transfer their content into various recipient cells inside the tissues (Figure 1). Different EV uptake mechanisms by the target cell have been described including membrane fusion, ligand binding interaction or EV endocytosis, as reviewed in [8]. These membranous vesicles are heterogeneous in size and have given rise to numerous names (exosomes, microvesicles, microparticles, prostasomes, oncosomes, neurospheres, apoptotic bodies, etc.). The growing interest in EVs and the recent advances in the characterization of their biogenesis pathways have led the scientific community to propose a nomenclature that essentially distinguishes two subtypes of EV based on their sizes: large and small EVs [10].

Large EVs (lEV), whose sizes vary between 250 nm to 500 nm in diameter and are mainly referred to as microvesicles, are secreted following the budding of the plasma membrane. This process is dependent on calcium influx, which induces modification of the asymmetric phospholipid distribution of plasma membranes and phosphatidylserine outer leaflet exposure through specific regulation of enzyme activities (flippase, floppase and scramblase) and favors the reorganization of cytoskeleton through calpain activation [11]. First considered as platelet wastes, they aroused particularly strong interest as diagnostic tools in pro-thrombotic diseases [12]. Small EV (sEV, 40–100nm), referred to as exosomes, are derived from intraluminal vesicles formed during the maturation of multivesicular bodies (MVB) in the endolysosomal pathway. They are secreted into the extracellular medium after fusion of MVB with the plasma membrane. Detailed mechanisms governing the biogenesis, secretion, targeting and fate of EVs have been reviewed elsewhere [8]. Several studies have demonstrated the ability of sEVs to regulate the immune and anti-tumor response, in particular due to their ability to transfer major histocompatibility complex (MHC) molecules between immune cells [13]. Despite different modes of biogenesis, lEV and sEV share many common characteristics such as their similar appearance, an overlapping size and common cargos, which makes it difficult to ascertain the respective origin and role of each EV subtype after purification. Whereas most studies refer to a mix of EVs, differential ultracentrifugation (dUC) is usually used to get rid of cell debris and to separate lEVs (10,000–20,000× *g* pellet) from sEVs (>100,000× *g* pellet). Regardless of the EVs’ subclasses, numerous recent data point out that EVs convey biological messages and are critical actors of intercellular communication [14], being involved in tissue development and homeostasis in physiological and pathological conditions.

## 3. Adipocytes Are Important EV Providers

EV release has been recently identified as an essential part of the WAT secretome participating in autocrine, paracrine and endocrine communication [15]. We have investigated the ability of 3T3-L1 adipocytes to secrete EVs and demonstrated that adipocytes release two subtypes of EVs (lEV and sEV) [16]. The lEV fraction includes a heterogeneous population of vesicles, with a well-delimited double membrane, differing greatly in size, shape and electron density, whereas the sEV fraction corresponds to a pool of smaller spherical vesicles of similar sizes, with cup-shaped morphologies usually observed for exosomes. Quantification of AdEV revealed the ability of adipocytes to secrete important quantities of large and small EVs, from either in vitro adipocytes models (3T3-L1 or 3T3F442A) or mice primary adipocytes [16,17,18]. When compared to melanoma cells, which are known to secrete many extracellular vesicles, mature adipocytes appear as important EV providers as they release more sEV than cancer cells [18], whereas AdEVs’ isolation rate can be significantly enhanced using size exclusion chomatography (SEC) by comparison to classical dUC technique [19]. However, the presence of the lipid droplet marker perilipin-1 and the high lipid content in SEC-isolated AdEV preparations suggest that a bias may reside in the co-isolation of lipid droplets and lipoproteins with AdEV.

### 3.1. Adipose-Derived EV: A Complex Network of Metabolic Signals Inside WAT

Studying EV production during the course of 3T3-L1 differentiation reveals that proliferative adipocytes secrete more EV than quiescent mature adipocytes [17]. Indeed, lipid-filling during the course of adipocyte differentiation is associated with enhanced EV secretion [18]. Accordingly, when large and small adipocytes are size-separated from the same fat pad, large adipocytes exhibit higher efficacy in releasing sEV harboring glycosylphosphatidylinositol (GPI)-anchored protein, CD73 compared to small adipocytes [20]. Besides adipocyte’s intrinsic ability to secrete AdEV, the pathophysiological environment also influences AdEV secretion. Based on in vitro experiments, it was found that saturated fatty acids [16,21,22], pro-inflammatory cytokines [16] and hypoxia [23] significantly enhanced adipocyte-derived EV secretion and modulated EV content. Of interest, the mentioned stimuli are all related to the pathophysiological state of obesity and are in agreement with experimental data showing that the number of sEVs shed by adipocytes from obese mice is higher than that from lean animals [18]. Interestingly, increased sEV secretion in obese conditions is not observed for other cell types found in WAT (referred to as “stromal vascular fraction” or SVF) [18] although the presence of SVF markers detected in WAT-derived EVs affects the ability of SVF to produce EV [24].

Active EV trafficking exists between different WAT cell types such as, for example, endothelial and adipocytes. For instance, endothelial EV-based transfer of caveolin into adipocytes has been demonstrated and shown to be sufficiently efficient to restore caveolin-1 protein levels in adipocytes depleted for caveolin-1 [24]. EV-based cross-talks have been documented between all cell types composing WAT (Figure 2). For instance, such AdEV traffic occurs between small and large adipocytes and participates in the different stages of fat development [20,23,25]. Another important EV-based dialogue is the one occurring between adipose stem cells and the immune or endothelial cells, which contribute to impact WAT properties by modulating inflammatory markers or by regulating vascularization of the tissue [24,26,27,28]. Finally, the most documented WAT EV-based communication in the literature is the one between adipocytes and immune cells, which regulates their respective phenotypes [19,29,30,31].

### 3.2. Contribution of Adipocyte-Derived EV to the Circulating Pool of EV in Biofluids

Evidence for the presence of AdEV in the blood is still scarce and difficult to evaluate in the absence of adipocyte-derived specific markers, but highly expressed adipocyte proteins such as aP2/FABP4, perilipin-1, adiponectin or PPAR have been identified in circulating EV [17]. Nonetheless, their use as specific adipocyte markers is limited by the fact that their expressions vary according to adipocyte differentiation and/or hypertrophy, and that most of these proteins are also expressed by WAT-derived macrophages. Using a fat-specific knockout of the miRNA-processing enzyme Dicer (ADicerKO), it was shown that fat was a major contributor to circulating exosomal miRNA [32]. Conversely, tracing plasma EV in mice expressing a fluorescent protein specifically in adipocytes revealed that AdEVs were indeed detected but represented a minority of circulating EV [19]. Further investigations will thus be needed in order to evaluate the exact contribution of AdEV in the circulating pool of EV with regard to the well-known predominance of platelet-derived EV and to a lesser extent to endothelium and PBMC-derived EVs [33].

### 3.3. Adipose-Derived EV Regulate Glucose Homeostasis and Inflammation

Obesity, and particularly adipocyte hypertrophy, is an important contributor to type 2 diabetes (T2D), with insulin resistance being the main hallmark [34]. Different studies have investigated the metabolic effects of obese WAT-derived EV on insulin signaling. Blood injections of sEV derived from obese WAT into lean mice altered insulin sensitivity and induced insulin resistance compared to injections of sEVs isolated from lean fat depots, illustrating the potential of WAT-derived EVs to act as metabolic perturbators [35]. One particular mechanism linking obese adipose sEV and insulin resistance relies on the ability of obese AdEVs to chemoattract monocytes, therefore contributing to WAT inflammation [36,37]. Accordingly, sEVs released from macrophages from obese WAT also caused systemic insulin resistance when administered into lean mice, suggesting an important contribution of macrophages-derived sEV in addition to AdEV in metabolic diseases [30]. Finally, EV-induced adipose remodeling might also result from inter-organ EV trafficking as recently illustrated by hepatocyte-derived EV targeting adipocytes to regulate adipogenesis and lipogenesis [38].

Besides the role attributed to AdEV in WAT homeostasis, some studies have highlighted the endocrine effects of AdEV on distant cells from other tissues. EVs from brown adipose tissue (BAT) have been shown to regulate gene expression in the liver, although no evidence for specific organotropism was demonstrated [32]. Proatherogenic properties of obese adipose-derived sEV, specifically when isolated from visceral depots vs. subcutaneous WAT, have been reported to be exerted by regulating macrophage foam cell formation and polarization [39]. Finally, the role of AdEV in tumor-WAT communication has also been demonstrated [18,40]. Metabolic changes could be horizontally induced in melanoma cells by AdEV, which resulted in the increase in tumor aggressiveness, tumor cell migration and lung metastases, which were reinforced in the context of obesity [18].

Taken together, these studies position adipose-derived EVs as a novel means of intercellular communication within WAT and likely between WAT and other distant organs. Of note, most of these studies have focused on the sEV and have neglected the lEV subpopulation. In addition, the risk of contamination of AdEV with lipoproteins contaminants and/or macromolecular protein complex is not considered or discussed. Further studies using standardized and robust EV isolation protocols are thus needed to delineate the molecular mechanisms underlying EV paracrine and endocrine effects.

## 4. Adipose-Derived EV Content Explains Their Biological Functions

Different biological functions of AdEVs have been identified, and specific AdEV components (proteins, lipids or acid nucleics) have been assigned to these effects.

### 4.1. Protein Content of Adipocyte-Derived Extracellular Vesicles

The formation of EVs implies that membranes can bud away from the cytoplasm during the formation of large EVs or during the formation of the intraluminal vesicles inside late endosomes to generate sEV/exosomes (Figure 1). During the budding process, membrane-associated proteins are incorporated into EVs and their nature is highly dependent on EV intracellular origins. EV-enclosed proteins can be those involved in the budding itself, like ESCRT machinery component for exosome/sEV formation (Endosomal Sorting Complexes Required for Transport [8]), or small GTPases (ARF62,18 and ARF119), Rab proteins and Rho (Rac1 and RhoA) for large vesicle formation [41]. Regarding proteins released into sEV, accumulating evidence suggests that post-translational modifications are necessary for their incorporation into sEV during MVB biogenesis [42]. In addition, since MVB are signaling platforms for many signaling pathways, scaffold proteins are also often retrieved into sEVs [43].

Therefore, when comparing 3T3-L1 adipocyte-derived EV proteomes, we highlighted specific protein subsets carried by lEV or sEV reflecting their respective mode of biogenesis [16]. Large EVs were enriched in plasma membrane proteins (including flotillin-1 and caveolin-1), organelle components and mitochondrial enzymes, whereas proteins from endosomal origin (including CD63 and CD9), from extracellular matrix or involved in cell adhesion were specifically retrieved in sEV. Of note, sEV proteins constitute an essential part of human WAT secretome [15]. As shown on Figure 3a, sEV proteins were specifically enriched in proteins for translation and RNA processing when compared with the subset of proteins released in a vesicle-free manner. Conversely, sEV proteins were depleted in proteins involved in immune response, which were enriched in the subset of vesicle-free proteins (Figure 3a). This analysis suggests that two pools of proteins with specific biological functions are released from adipocytes, and identifies sEV as a specific sorting pathway for proteins involved in RNA processing/translation. Surprisingly, 36% of the proteins found in human adipocyte-derived sEVs had conventional secretory signal peptides [15], suggesting either that these proteins are contaminants attached to AdEVs being precipitated during EV extraction, or that they are incorporated into MVBs during their intracellular trafficking export. In line with this second hypothesis, during lipolytic stimulation associated with lipid mobilization, FABP4, a lipid transporter usually exported under a free form in the plasma, is recruited into MVB and released into AdEV [44].

In human blood, different adipocyte-enriched proteins or adipokines are still detected in circulating EVs even after post-depletion of the major population of circulating plasma-derived EV (including platelet, monocyte, endothelial cell and erythrocyte-derived EVs) [17]. These proteins, including adiponectin, FABP4, perilipin and PPARγ, could therefore use an adipocyte-EV secretory pathway to be exported in blood. Nonetheless, previous data demonstrated that adiponectin is mainly distributed at the exosomal surface, whereas adiponectin is not usually membrane-associated [47]. This raises the question of the proportion of co-precipitated or EV-adsorbed soluble material when evaluating adipocyte protein marker content in circulating EV. In addition, sequential depletion of lEV then sEV from blood patients has a limited impact on most circulating adipokine concentrations, demonstrating that the EV secretory pathway remains negligible for most of these proteins [33]. An exception is the Macrophage Migration Inhibitory Factor (MIF), whose lEV transport accounts for half of its circulating concentration, a secretory pathway that, moreover, is conserved over different MIF-producing cells [33]. Large EV-associated MIF triggers rapid ERK1/2 activation in macrophages, and these functional lEV-MIF effects specifically rely on a non-canonical MIF tautomerase activity. Altogether, these results highlight that a specific adipokine sorting pathway does exist, besides their default encapsulation in EV, which would need to be reconsidered when studying metabolic effects of adipocyte secreted products.

Cellular origin is also likely to influence EV protein content. To illustrate the relative contribution of EV origin with regard to their EV protein content, we performed functional enrichment analyses of two different EV sets of proteins, respectively derived from subcutaneous human primary adipocytes [15] and human skeletal muscle cells (SkM) [45]. Figure 3b shows that human AdEVs vs. SkMEVs display different protein signatures. SkMEVs appeared to be enriched in proteins involved in neuromuscular development and cell differentiation, whereas AdEVs were significantly enriched in proteins involved in RNA processing and translation and in proteins from the extracellular matrix (ECM). Interestingly, proteomic analysis of murine 3T3-L1 adipocyte-derived sEV also demonstrated a significant number of ECM proteins in AdEV [16], as illustrated for AdEV-associated MMP-3, which can even be transferred into lung cancer cells [40]. Such sEV protein composition could reflect the important production and organization of ECM associated with WAT development [48,49].

AdEV release is also closely linked with the adipocyte developmental stages since sEV production from 3T3-L1 cells was greater for the pre-adipocyte stage than for mature adipocytes [50]. A plausible explanation is the saturation of the ubiquitin–proteasome or autophagy–lysosomal pathways due to high protein turnover associated to with proliferation, which favors the release of cellular toxic components by EV from preadipocytes [51]. In addition, AdEV composition varies during 3T3-L1 adipocyte differentiation [50]. Bioinformatics analyses of the proteomic signatures of pre- vs. differentiated adipocyte sEVs by using the same procedure described in Figure 3 indicate that pre-adipocyte-derived sEV are significantly enriched in proteins for angiogenesis (*p*-value: 3.77 × 10^−10^) and positive regulation of cell motility (*p*-value: 6.04 × 10^−10^). Conversely, differentiated 3T3-L1 adipocyte-sEV are enriched in proteins for RNA catabolic process (*p*-value: 1.38 × 10^−5^), cell cycle phase (*p*-value: 6.09 × 10^−4^), mitotic cell cycle phase (*p*-value: 6.09 × 10^−4^), post-translational protein modification (*p*-value: 9.30 × 10^−5^) and immune response-activating signal transduction (*p*-value: 6.09 × 10^−4^).

AdEV production is also dependent on environmental signals, especially those associated with the development of obesity, as illustrated by an increase in AdEV release following lipid/glucose, hypoxic or inflammatory stimuli [16,22,23,37]. Metabolic alterations, such as insulin-resistance and lipid hypertrophy (induced by oleate or palmitate treatment) applied to murine cell-cultured adipocytes C3H10T1/2, moreover impacts the AdEV protein content [22]. Lipid hypertrophied AdEVs are characterized by ceruloplasmin, mimecan and perilipin 1 adipokines, and those from the insulin-resistant adipocytes by the striking presence of the transforming growth factor-beta-induced protein ig-h3 (TFGBI). AdEV cargo differential contents are likely to modulate metabolic responses of recipient cells. For instance, “hypoxic” AdEVs affect lipogenic activity in neighboring pre-adipocytes and adipocytes [23] and alter the insulin-stimulated signaling pathway [25]. AdEVs derived from hypertrophic adipocytes, following oleic acid or palmitate treatments, recapitulated differentiation/hypertrophy and induced insulin resistance in recipient adipocytes or promoted macrophage inflammation by stimulating IL-6 and TNFalpha expressions [22]. Finally, “inflammatory” AdEVs, produced from adipocytes treated with TNFalpha and co-exposed or not to hypoxia, induce VCAM-1 production in vascular endothelial cells, resulting in enhanced leukocyte attachment [37].

Proteomic studies performed on AdEV from WAT explants of lean or obese/diabetic rodents (genetic obesity or high-fat induced) confirmed alterations of AdEV protein content within the course of obesity [22,35,52,53]. Following a comparison of these three proteomic analyses, we found that AdEVs from high-fat-diet-induced obese animals (obese AdEVs) generally display higher content of proteins compared with AdEV from control rodents (Figure 4). Focusing on the proteins commonly found in obese AdEVs vs. control AdEV, a subset of 65 proteins could be identified (see the list in Table 1). Functional enrichment analysis indicated that the 65 proteins are involved in lipid catabolic processes and oxydo-reduction, cell migration and motility and were located in caveolae and extracellular matrix.

Alternatively, a recent proteomic analysis was performed on human morbid obese visceral (VAT) and subcutaneous (SAT) WAT shed EVs from donors submitted to bariatric surgery [54]. Functional analysis of all the proteins identified in obese VAT and SAT vesicles showed the presence of proteins related to transport, catalytic, GTPase, structural molecule, protease and chaperone activity and a particular enrichment of extracellular matrix (ECM) constituents in SAT EVs. Importantly, the vast majority of proteins identified in previous proteomic reports from EVs derived from cultured adipocytes [15,16,17,18,22] were retrieved in human WAT EVs [54]. Other proteins, including leptin, have not been previously described in AdEV from in vitro adipocyte differentiated cultured models, which are also known to be low producers of this adipokine. The functional classification shows that obese VAT vesicles display a specific enrichment of proteins implicated in WAT inflammation and insulin resistance, related to a specific increase in protein implicated in the immune system process, in comparison to SAT EVs [54]. Since obese VAT is recognized to be more inflamed than SAT due to important macrophage and immune cell infiltration, EVs derived from WAT-resident immune cell populations are likely to impact WAT-EV dynamic secretion and protein content. Finally, the authors revealed a particular enrichment of human obese WAT EVs in TGFBI and mimecan, two proteins that they also found associated with plasma EVs from obese patients [54]. Interestingly, plasma EV-associated TGFBI was significantly elevated in obese patients with a history of T2D compared to non-diabetic patients, and mimecan-EVs were higher in obese plasma compared to those in healthy lean individuals and may therefore represent candidate biomarkers to monitor T2D status in obese patients or to track obesity, respectively. However, one must be conscious that these two proteins are not exclusively secreted by WAT, and other cell types than adipocytes are likely to participate to increase their plasma TGFBI-EV or mimecan-EV levels.

Of interest, AdEV proteins could be transferred into various recipient cells and are likely to participate in cancer development [52], inflammation and insulin resistance development [35]. Indeed, AdEV stimulated mitochondrial metabolism and remodeling in tumor cells by providing both enzymes and substrates [52]. Alternatively, obese AdEVs were found to contain higher levels of the RBP4 protein compared to lean WAT-derived AdEV, involved in M1 macrophage polarization and insulin resistance in a TLR4/TRIF-dependent pathway [35]. Whether AdEV’s deleterious metabolic effects operate via AdEV protein delivery into recipient cells or also involve AdEV indirect mechanisms, as illustrated by their TLR4-dependent immuno-modulatory effects [35], will definitely need further investigations.

### 4.2. RNA in Adipocyte-Derived Extracellular Vesicles

RNAs have been consistently found in EVs. Until now, the mechanisms favoring their export into EVs is unclear and unexplored in the case of AdEVs. Generally speaking, like for the majority of RNA-associated with EV, AdEV-RNA concentrations mirror cellular intracellular concentrations, suggesting a passive mechanism [55]. Nonetheless, for some small RNAs, different mechanisms underpin this EV-associated RNA sorting, which are not mutually exclusive, including (i) specific RNA sequences with affinity for raft-like region of MVB (review in [56]); (ii) binding to specific RNA-binding proteins that selectively shuttle miRNA into EV (review in [57]); (iii) the presence of specific acid nucleic extension, which might stabilize some miRNAs and favor their export [58]. The majority of sEV mRNA is fragmented, which may participate in their stability, localization and mRNA translational repression in recipient cells [59].

Only two studies have performed large-scale analyses of AdEV RNA content. Microarray profiling identified 7000 mRNA in 3T3-L1 adipocytes among the 9000 expressed in the cell [55]. This high number of AdEV mRNA is quite surprising given the fact that the authors indicated that the majority of RNA in AdEV were less than 200 nucleotides in length and contained little or no 28S and 18S ribosomal RNA compared to the parental adipocytes. Adipocyte-specific transcripts were identified coding for adiponectin, leptin, resistin, PPARgamma, FABP4, C/EBPs [55]. RAW264.7 macrophages incubated with AdEV expressed these adipocytes-specific transcripts, suggesting that mRNA can be transported into macrophages through the AdEV route [55]. These data corroborated the study of Müller et al. showing that AdEV from large adipocytes transfer transcripts coding for fatty acid esterification (glycerol-3-phosphate acyltransferase-3, diacylglycerol acyltransferase-2), lipid droplet biogenesis (FSP27, caveolin-1) and adipokines (leptin, adiponectin) into small adipocytes and that such RNA horizontal transfer correlates with the induction of lipid storage in the recipient cells [60]. By using RNA sequencing, 1083 mRNAs and 105 lncRNA were moreover identified in AdEV from bovine adipocytes out of the 12,082 mRNAs and 8589 lncRNA expressed in donor adipocytes, therefore confirming the presence of long RNA species in AdEV. Respectively, 498 mRNA and 68 lncRNA were found differentially expressed between adipocytes and AdEV [61]. The 500 highly concentrated mRNA in AdEV coded for proteins involved in translation, protein folding and collagen fibril organization, or ribosome and cytoskeleton proteins. Interestingly, like for the protein content of AdEV, the extracellular matrix was among the enriched functions. Among the 105 lncRNAs, 3 lncRNAs (BGIR9913_49345, BGIR9913_54344 and URS0000B2F7C9) were detected in sEV irrespective of cellular origin suggesting a conserved mechanism for their upload into sEV. Until now, the functionality of AdEV mRNA, i.e., their translation into proteins in the recipient cells, has not been demonstrated. However, a previous study has shown that after incubation of human mast cells with mouse EV mRNA, new mouse proteins were found in human recipient cells, demonstrating that transferred mRNA could be translated into proteins in other cells [14]. In addition to mRNA and lncRNAs, it was found that AdEV also contained circular RNA [62]. Circular RNA functions as a sponge for miRNAs expressed in the recipient cells. They can also regulate RNA-binding proteins and can sometimes be translated into proteins. Circular RNAs contained in AdEV promoted hepatocellular carcinoma growth and reduced DNA damage by suppressing miR-34a, resulting in the activation of the USP7/Cyclin A2 signaling pathway [62]. AdEVs from adipocytes overexpressing circ_0075932 were enriched in circ_0075932 and induced inflammation and apoptosis in dermal keratinocytes. It was demonstrated that circ_0075932 binds the RNA-binding protein PUM2, a positive regulator of the AuroraA kinase, resulting in the activation the NF-κB pathway.

AdEVs also contain small RNA species. Out of the 378 miRNAs expressed in bovine adipocytes, 48 were sorted into AdEVs [61] and 140 were also found in 3T3-L1-released AdEVs [55]. Different pieces of evidence from in vitro data highlight AdEV-miRNA horizontal transfer into various recipient cells. For instance, hypertrophic adipocytes released AdEVs enriched in miR-802-5p, which contributed to insulin resistance in cardiac myocytes through its action on HSP60 [63]. miR-27a contained in AdEV derived from high-fat-diet-fed C57BL/6J mice induced insulin resistance in C2C12 skeletal muscle cells by repressing PPARγ and its downstream genes [64]. In addition to muscle cells, AdEVs were also implicated in the cross-talk between adipocytes and the liver. Thomou et al. found that miR-99b in AdEV reduced Fgf21 mRNA levels in the liver and demonstrated that FGF21 modulation only occurred through AdEV delivery of miR-99b and not in response to direct incubation with miR-99b [32]. This result has suggested, for the first time, a specific role of packed miRNAs vs. vesicle-free miRNAs in blood. AdEVs were also implicated in hepatic cancer development, as the transport of miRNA 23a/b into hepatic cancer cells via AdEV resulted in cancer cell growth and migration and development of chemoresistance through targeting of the von Hippel-Lindau/hypoxia-inducible factor axis [65]. Within WAT, it was demonstrated that AdEV could participate in macrophage polarization. Zhang et al. showed that miR-155 could be delivered into bone-marrow-derived macrophages by AdEV, which resulted in the targeting of SOCS1 and the modulation of M1 macrophage polarization via JAK/STAT signaling [66]. Interestingly, the conditioned medium of macrophages pre-stimulated with miR-155-bearing AdEV regulated insulin signaling and glucose uptake in adipocytes. Additionally, AdEV-released miR-34a could be transported into macrophages, resulting in the inhibition of M2 polarization through inhibition of the expression of Krüppel-like factor 4 [67]. Together, these data illustrate the complex interplay between adipocytes and macrophages, which can partly be explained by the exchange of vesicle-packed miRNA.

### 4.3. Lipids in AdEV

Although EVs are membrane-derived vesicles, one often-neglected component is their lipid content that EVs also transfer into recipient cells. EVs display specific lipid enrichment. Their membrane high protein/lipid ratio and the lipid asymetric distribution confer a high membrane rigidity in comparison with parent cells, which explains their stability in biofluids [68]. Interestingly, it was demonstrated that EV protein and lipid enrichment mechanisms are not linked. Indeed, some cell types differing in protein and lipid composition secrete EV enriched in the same subgroup of proteins but not the same species of lipids. Conversely, EV lipid content might reflect the lipid composition of their parental cells, whereas the EV proteins differed [69]. These data strongly suggest that combining lipid and proteomic profiles from EVs could help better define specific AdEV biomarkers.

Several studies on cancer cells have shown that sEVs are strongly enriched in cholesterol, sphingomyelin (SM), glycosphingolipids and phosphatidylserine (PS) (mol% of total lipids) and depleted in phosphatidylcholine (PC) (see review in [70]). Interestingly, compared to these cancer-derived EV, AdEVs have a different lipid distribution and enrichment from the parental cells. Indeed, in both sEVs and lEVs released from 3T3-L1 adipocytes, PC represents by far the main phospholipids, whereas LysoPC, PS, phosphatidylinositol (PI) and phosphatidylethanolamine (PE) are proportionally minor phospholipids [16,50]. In addition, sEV and lEV lipid compositions relate to their mode of biogenesis [69]. For instance, whereas 3T3-L1-derived sEV and lEV display similar phospholipid profiles, sEVs have a specific cholesterol enrichment known to be a trait of exosomes acquired during their biogenesis, whereas a high amount of externalized PS is retrieved in lEV in line with the pro-coagulant potential of this lEV subclass [16].

Lipidomic analyses from AdEV are scarce, and the role of AdEV lipids in their biological functions in recipient cells needs urgently to be determined. During the development of obesity, important membrane remodeling occurs which is also illustrated by plasma membrane lipids reorganization. Of note, adipocyte plasma membrane lipids, such as cholesterol or sphingomyelin concentrations, are closely linked with the development of obesity-associated metabolic complications including insulin resistance [71,72]. Therefore, as a consequence, AdEV lipid composition might also be affected and could modulate some biological functions into the recipient cells. In line with this hypothesis, AdEV released from pre-differentiated or post-differentiated 3T3-L1 adipocytes displayed a different phospholipid composition closely resembling the phospholipid composition of the parental adipocytes, especially for PE and PS, confirming that modifications of adipocyte lipid composition could be reflected in AdEV [50]. Interestingly, Clement et al. demonstrated that AdEV free fatty acids could be taken up by melanoma cells stimulating fatty acid oxidation and melanoma migration [52]. Although this study did not indicate whether the lipid composition of AdEV also participated in melanoma aggressiveness, it demonstrated for the first time that AdEV could spread lipids in other tissues/cell types. Such AdEV lipid sorting has even been proposed as a second pathway of lipid release from adipocytes that is independent of the canonical lipolysis and that feeds local macrophages with AdEV lipids [19]. The authors estimated that WAT from lean mice may release ~1% of its lipid content per day via AdEVs ex vivo, a rate that is more than doubled in obese animals. Nonetheless, this percentage could be overestimated by a co-isolation of adipocytes with contaminant lipid droplets. The lipid class particularly enriched in sEVs is ceramides, which are also deleterious lipids interfering with insulin sensitivity in insulin-sensitive tissues [73]. AdEVs derived from WAT explants presented a specific signature in ceramides and displayed high levels of sphinganine, sphingosine-1 phosphate (S1P) and all sphingomyelin species, which are likely to alter a wide range of signaling pathways within WAT [24].

In the context of obesity, AdEVs released from adipocyte explants from high-fat-diet obese mice are strongly enriched in palmitic and stearic acids by comparison to AdEVs from standard diet mice, suggesting that the quality of the diet also has an impact on AdEV lipid composition [35]. For instance, a diet specifically enriched in palmitate triggered the release of EVs highly enriched in palmitate from skeletal muscle and changed their biological properties and perturbed skeletal muscle homeostasis [74].

## 5. Therapeutic Strategies to Decrease Ad EV Deleterious Effects

As AdEVs appear as important metabolic mediators in obesity-associated pathologies, designing EV-based strategies to counteract deleterious AdEV effects, particularly in the pathophysiological context of obesity, might be envisaged. However, such approaches would imply specifically targetting AdEVs.

### 5.1. Targeting AdEV Extracellular Vesicle Biogenesis and Release

In order to counteract AdEVs’ biological effects, one strategy to be envisaged might consist in modulating AdEV formation. Many drugs targeting either sEV formation or budding of the plasma membrane have been tested with promising results, mainly in the treatment of cancers (for a review, see [75]). Very interestingly, it seems that some of the tested inhibitors are known to regulate proteins involved in the development of insulinresistance in adipocytes and thus their use to restore insulin-sensitivity might also be a therapeutic strategy to reduce EV release from WAT. For instance, calpeptin, a cystein proteinase inhibitor, can be used to target calpains involved in lEV production. It has been shown that calpain inhibition attenuated WAT inflammation and suppressed macrophages migration to adipose tissue in vitro [76]. In addition, as calpain inhibition restores autophagy [77], it could favor the targeting of MVB to the autolysosome pathway for ILV degradation, resulting in a decrease in sEV sorting [78]. Another interesting drug is the anti-hypertensive Y27632 compound that targets RhoA-Rho kinase ROCK1/2 proteins involved in lEV formation [79,80]. Over-activation of the ROCK pathway has been implicated in the development of adipocyte hypertrophy, in the increase in inflammatory cytokine production and in the development of obesity-induced insulin resistance [81]. As partial deletion of ROCK1 or ROCK2 has been found to attenuate high-fat-diet obesity-induced insulin resistance [81,82], the use of Y27632 in patients suffering from obesity could be a strategy to decrease AdEV release and reduce cardiovascular diseases associated with obesity [83].

Besides modulating EV proteins involved in MVB biogenesis and lEV budding, the regulation of specific intracellular lipid concentrations could be also envisaged for EV production. Indeed, an alternative ESCRT-independent pathway for EV biogenesis has been described involving the generation of ceramides. Ceramides are cone-shaped lipids that can both induce inward budding of MVB to generate ILVs and the release of sEV, and plasma membrane budding to generate lEVs (Figure 1), as they preferentially accumulate in the inner membranes creating lipid-raft domains. The increased concentration of ceramides in tissues is associated with the consumption of high-saturated fatty acids diets and/or are induced by inflammatory cytokines. In this context, it might be interesting to test whether the drug GW4869, which can decrease the generation of sEV from cells through its action on the membrane neutral sphingomyelinase (nSMase) [84], could restore insulin-sensitivity in obese patients. In line with this suggestion, GW4869 has been shown to regulate inflammatory responses driven by TNFalpha from monocytes/macrophages [85]. In addition, inhibition of 3T3-L1 AdEV biogenesis and release following treatment with GW4869 could inhibit lipolysis and WAT browning, illustrating that such a strategy may be also useful for treating cancer-associated cachexia, a disorder characterized by unintended weight loss due to both skeletal muscle wasting and fat loss [86].

In addition to their involvement in EV release, lipids participate in the biological functions of EVs [74]. Therefore, modulation of AdEV lipid composition might beneficially modulate AdEV functions. In line with this suggestion, it was demonstrated that pharmacological inhibition of sphingosine kinase 1 (S1P1) in hepatocytes resulted in a significant reduction in S1P1-EV cargoes. Deleted-S1P1 EV decreased the migration responses of macrophages and consequently ameliorated non-alcoholic steatohepatitis [87]. Interestingly, pharmacological inhibition of sphingosine kinases 1 was shown to reverse obesity-inflammation in skeletal muscles of obese mice [88] and to reduce pancreatic lesions in spontaneously diabetic rats [89], therefore legitimating the use of such approaches to modulate both EV lipid composition and obesity-related disorders.

It has to be mentioned, however, that MVB trafficking and sEV/lEV release are parts of a complex intracellular trafficking and signaling networks, in close relationship with other cellular organelles to maintain cellular homeostasis and to release toxic components from the cells. Therefore, the full abortion of AdEV release cannot be envisaged as it would induce apoptosis. In addition, the question of targeting specifically adipose ceramide production in vivo fully remains speculative considering that adipocytes may not be the primary source of ceramides in WAT, which can rather be produced by other SVF cells.

### 5.2. Modulation of AdEV Lipid Composition by the Diet

An alternative strategy to modulate AdEV lipid content is to modulate the intracellular lipid composition of the donor cells. Previous studies found a palmitate enrichment in AdEV, as well as other deleterious lipids, when AdEVs were isolated from mice fed with high-saturated-fat diets [35]. A supplementation in omega-3 polyunsaturated fatty acids at the expense of omega-6 ones has beneficial effects on WAT and increases the production of omega-3 metabolites, thereby exerting positive metabolic effects (for review, see [90]). Therefore, a diet enriched in polyunsaturated fats and low trans fat would impact adipocyte lipid content, and consequently AdEV lipid composition, enhancing their beneficial properties. In line with this suggestion, it has been demonstrated that dietary protein restriction modifies the protein composition of circulating EVs, demonstrating that diet can directly impact EV composition [91].

It is well admitted that insulin-resistance associated with obesity increases the risk of cholesterol synthesis and release by the liver and its accumulation in WAT, leading to adipocyte hypertrophy. It was recently demonstrated that cholesterol from the diet can participate in this alteration [92]. For instance, cholesterol from MVB membrane could influence the fate of EV: on the one hand, lowering intracellular cholesterol level redirects MVB to lysosome degradation [93], and on the other hand, high cholesterol level is associated with an increase in EV biogenesis, release and uptake. These data illustrate that hypertrophic AdEV might disseminate cholesterol among WAT during the consumption of high-cholesterol diet and/or during the development of metabolic syndrome [94]. They also suggest that part of the action of the statins used to lower blood cholesterol level by regulating its synthesis in the liver might rely on both the reduction of blood liver-derived EV and on the decrease of AdEV production.

Indirect diet-effects to restore WAT function may also be envisaged and could possibly contribute to modulating AdEV content in a healthy manner. For instance, the gut microbiota is now recognized as a key component in the development of obesity and related metabolic complications. Evidence from animal studies and human clinical trials has suggested beneficial effects from prebiotic and various probiotic strains on physical, biochemical and metabolic parameters related to obesity [95]. Therefore, prebiotic or probiotic supplementations might participate in the improvement of WAT homeostasis by promoting AdEV beneficial contents and favorable metabolic effects. Alternatively, supplementation with EV from external sources might also contribute to restoring obesity-related WAT function and thereby modulate AdEV biological functions. For instance, we demonstrated that nanovesicles from orange juice could reverse high-fat-diet-induced gut modifications (e.g., length of villi and immune response) in diet-induced obese mice [96]. Additional investigations will be required to envisage diet and/or supplements as a strategy to modulate AdEV content in order to counteract their deleterious biological effects.

### 5.3. Use of Extracellular Vesicles from Healthy Subjects

Recent data have provided proofs of concept that EVs from healthy/young subjects might be used in the management of metabolic complications associated with obesity such as insulin-resistance or in the management of aging-associated metabolic disorders. Indeed, injections of WAT macrophage-derived EVs isolated from lean mice improved glucose tolerance and insulin sensitivity in diet-induced obese mice [30]. Similarly, injection of young (3-month-old) mice blood EVs into aged (18-month-old) mice reversed the expression of aging-derived biomarkers [97]. Other studies have evidenced that EV isolated from adipose-derived stem cells from healthy patients displayed cardiac regenerative properties [98], or could improve insulin sensitivity, reduced obesity, and alleviated hepatic steatosis in diet-induced obese mice by reducing inflammation [28]. Finally, some studies have suggested that many of the “exerkines” are contained within circulating EVs and might participate in the beneficial effects of exercise on obesity and type 2 diabetes (for review, [99]). Together, these studies suggest that the use of EVs derived from “healthy” WAT might be a potential strategy in addition to a modification of lifestyle and the use of drugs to normalize glycemia, and this would deserve to be investigated.

### 5.4. Use of Antibodies against AdEV

In the context of cancer, it has been demonstrated that blood injections of antibodies against CD63 and/or CD9, two tetraspanins expressed at the surface of all EVs, could significantly reduce the development of metastasis without any effects on tumor growth in mice [100]. As these two antibodies were not specific to the tumor-derived EVs, the authors explained this result by a general decrease in EV flux between organs, including the tumor, and demonstrated that these antibodies stimulated the uptake of EVs by patrolling macrophages. It was also demonstrated that the use sof a fragment of CD9-antibody could prevent the transfer of tumor-derived EV cargoes in recipient cells in vitro [101]. We previously demonstrated that obese patients have higher levels of circulating EVs in comparison to healthy patients [100], suggesting increased deleterious cross-talk between metabolic organs, including WAT. Therefore, the strategy to use antibodies against EV to reduce the EV flux in obese patients could be a complementary strategy during weight loss.

## 6. Conclusions

In this review, we provide evidence that extracellular vesicles released from adipocytes (AdEVs) participate in the homeostasis of adipose tissue by exchanging lipids, proteins and RNA between the different cells that compose the fat tissue. AdEV composition is closely connected to the composition of the secretory cells, and the pathophysiological context of obesity impacts EV content. AdEVs thereby participate in the instigation of inflammation and insulin resistance of adipose tissue and are also involved in the spread of cancer cells. Nonetheless, numerous questions remain unanswered. They will need to be resolved in the future prior to envisaging therapeutic avenues to counteract the deleterious effect of AdEV during the development of obesity.

## Figures and Tables

**Figure 1 ijms-22-01788-f001:**
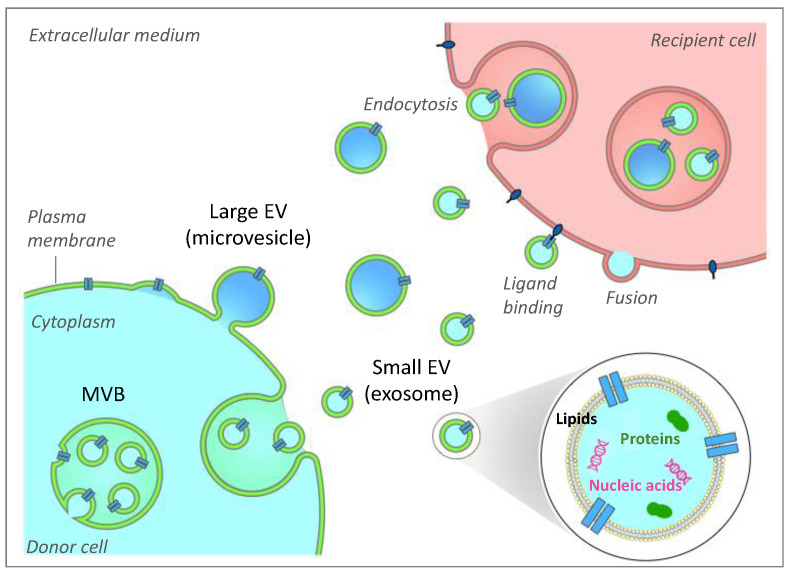
Extracellular vesicle biogenesis, secretion and interaction with recipient cells. Two subclasses of extracellular vesicles (EV) are released from mammalian cells and are mainly distinguished based on their sizes. Large EVs (lEV) can bud from the plasma membrane. They are also referred to as microvesicles. Small EVs are derived from the formation of intraluminal vesicles (ILVs) within the lumen of the multivesicular body (MVB). MVB can fuse with the plasma membrane to release ILVs, which are thus called exosomes. EVs participate in intercellular communication through EV-based exchanges of proteins, lipids and genetic material between cells. The fate of EVs in recipient cells includes membrane fusion, ligand binding or endocytosis mechanisms.

**Figure 2 ijms-22-01788-f002:**
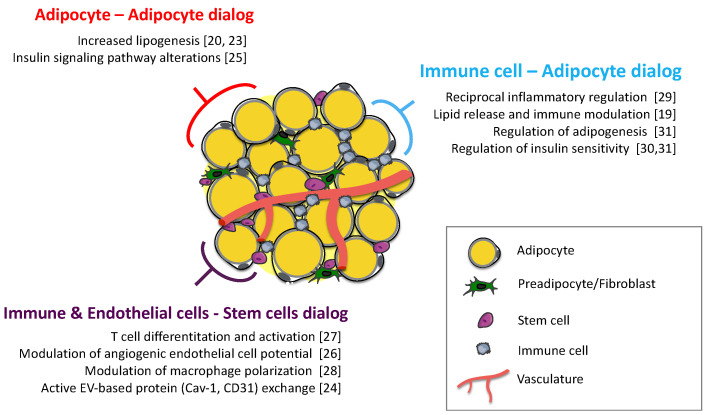
Cellular EV-based cross-talks between adipose tissue modulate fat metabolism by autocrine action. EV-based dialog within white adipose tissue (WAT) has been evidenced between the different fat cell types as illustrated between adipocytes themselves, or between stem cells/immune cells and endothelial cells and between immune cells and adipocytes. All these exchanges result in modulating cell recipients’ metabolic responses and participate in maintaining fat homeostasis. The references of the publications cited as examples are given in brackets.

**Figure 3 ijms-22-01788-f003:**
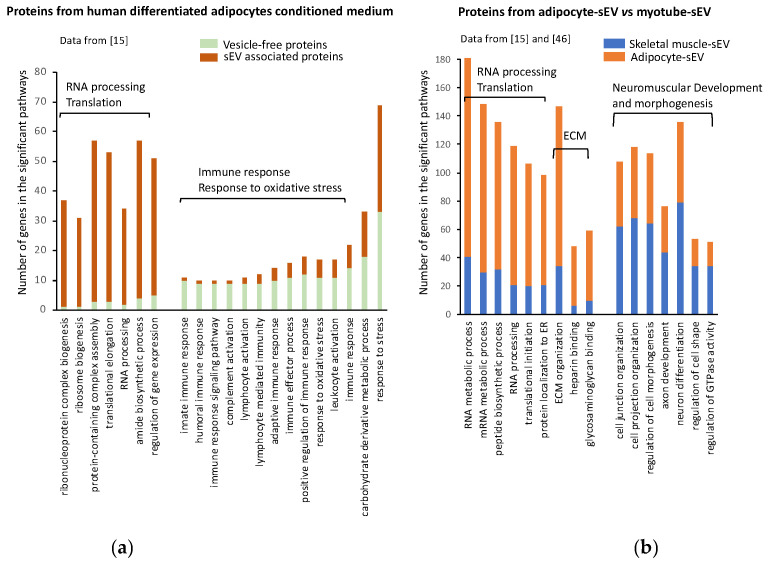
Adipocytes export a specific subpopulation of proteins into adipocyte-derived extracellular vesicles (AdEVs). (**a**) Significant GO biological functions enriched in proteins released in the conditioned medium from human differentiated adipocytes, either packed inside small extracellular vesicles or being released under a vesicle-free form. Proteomic data are from [15]. (**b**) Significant GO biological functions enriched in small EV (sEV) protein subsets released from human adipocytes vs. those released from human myotubes. Proteomic data are from [15] and [45]. Only functions containing more than 50 genes are presented. ECM, extracellular matrix; ER, endoplasmic reticulum. For (**a**) and (**b**), the significantly enriched pathways were retrieved using PANTHER version 11 [46].

**Figure 4 ijms-22-01788-f004:**
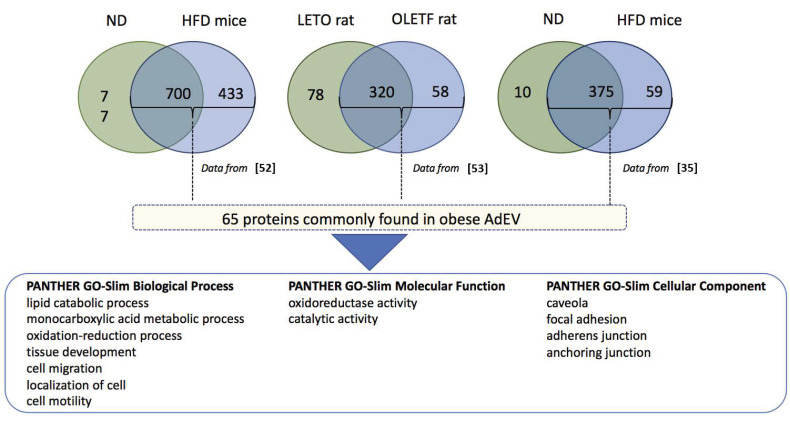
Significant enrichment analyses were performed on the 65 proteins commonly found in AdEVs released from obese mice adipose tissue based on proteomic data from [35,52,53] vs. the rest of the genome (PANTHER version 11). Only significant functions in each pathway are shown. The list of 65 proteins is available in Table 1.

**Table 1 ijms-22-01788-t001:** Common proteins from obese adipocyte-derived extracellular vesicles (see Figure 4). Data sets are from [35,52,53].

Gene Symbols	Protein Accession Numbers	Gene Names
Acadl	P51174	acyl-Coenzyme A dehydrogenase, long-chain
Acads	Q07417	acyl-Coenzyme A dehydrogenase, short chain
Aco2	Q99KI0	aconitase 2, mitochondrial
Acsl1	P41216	acyl-CoA synthetase long-chain family member 1
Adipoq	Q60994	adiponectin, C1Q and collagen domain containing
Agpat2	Q8K3K7	1-acylglycerol-3-phosphate O-acyltransferase 2 (lysophosphatidic acid acyltransferase, beta)
Aifm2	Q8BUE4	apoptosis-inducing factor, mitochondrion-associated 2
Aldh2	P47738	aldehyde dehydrogenase 2, mitochondrial
Aldh3a2	P47740	aldehyde dehydrogenase family 3, subfamily A2
Anxa1	P10107	annexin A1
Anxa6	P14824	annexin A6
Aoc3	O70423	amine oxidase, copper containing 3
Atp2a2	O55143	ATPase, Ca++ transporting, cardiac muscle, slow twitch 2
Atp5a1	Q03265	ATP synthase, H+ transporting, mitochondrial F1 complex, alpha subunit, isoform 1
Atp5b	P56480	ATP synthase, H+ transporting mitochondrial F1 complex, beta subunit
Cat	P24270	catalase
Cav1	P49817	caveolin, caveolae protein 1
Cav2	Q9WVC3	caveolin 2
Cct3	P80318	chaperonin subunit 3 (gamma)
Cd36	Q08857	CD36 antigen
Cd47	Q61735	CD47 antigen (Rh-related antigen, integrin-associated signal transducer)
Cd9	P40240	CD9 antigen
Cltc	Q68FD5	clathrin, heavy polypeptide (Hc)
Col6a1	Q04857	collagen, type VI, alpha 1
Decr1	Q9CQ62	2,4-dienoyl CoA reductase 1, mitochondrial
Dlat	Q8BMF4	dihydrolipoamide S-acetyltransferase (E2 component of pyruvate dehydrogenase complex)
Eef1a1	P10126	eukaryotic translation elongation factor 1 alpha 1
Ehd2	Q8BH64	EH-domain containing 2
Etfa	Q99LC5	electron transferring flavoprotein, alpha polypeptide
Fasn	P19096	fatty acid synthase
Gnaq	P21279	guanine nucleotide binding protein, alpha q polypeptide
Gpd1	P13707	glycerol-3-phosphate dehydrogenase 1 (soluble)
Gpi	P06745	glucose phosphate isomerase 1
Hadh	Q61425	hydroxyacyl-Coenzyme A dehydrogenase
Hadhb	Q99JY0	hydroxyacyl-Coenzyme A dehydrogenase/3-ketoacyl- Coenzyme A thiolase/enoyl-Coenzyme A hydratase (trifunctional protein), beta subunit
Hsd17b12	O70503	hydroxysteroid (17-beta) dehydrogenase 12
Hsd17b4	P51660	hydroxysteroid (17-beta) dehydrogenase 4
Itgb1	P09055	integrin beta 1 (fibronectin receptor beta)
Kpnb1	P70168	karyopherin (importin) beta 1
Lamb2	Q61292	laminin, beta 2
Lamc1	P02468	laminin, gamma 1
Ldha	P06151	lactate dehydrogenase A
Lipe	P54310	lipase, hormone sensitive
Lpcat3	Q91V01	membrane bound O-acyltransferase domain containing 5
Lpl	P11152	lipoprotein lipase; similar to Lipoprotein lipase precursor (LPL)
Lrp1	Q91ZX7	low density lipoprotein receptor-related protein 1
Mcam	Q8R2Y2	melanoma cell adhesion molecule
Mdh2	P08249	malate dehydrogenase 2, NAD (mitochondrial)
Ogdh	Q60597	oxoglutarate dehydrogenase (lipoamide)
Pc	Q05920	pyruvate carboxylase
Pdhb	Q9D051	pyruvate dehydrogenase (lipoamide) beta
Pdia3	P27773	protein disulfide isomerase associated 3
Phb	P67778	prohibitin
Prkar2b	P31324	protein kinase, cAMP dependent regulatory, type II beta
Rab18	P35293	RAB18, member RAS oncogene family
Rab8b	P61028	RAB8B, member RAS oncogene family
Rras	P10833	Harvey rat sarcoma oncogene, subgroup R
Sdha	Q8K2B3	succinate dehydrogenase complex, subunit A, flavoprotein (Fp)
Sfxn1	Q99JR1	sideroflexin 1
Sts	P50427	steroid sulfatase
Tmed10	Q9D1D4	transmembrane emp24-like trafficking protein 10 (yeast)
Tubb3	Q9ERD7	tubulin, beta 3

## Data Availability

No new data were created or analyzed in this study. Data sharing is not applicable to this article.

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
