# Peer review of "Adipocyte-Derived Extracellular Vesicles: State of the Art"

_ijms, 2021, doi:10.3390/ijms22041788_

Round 1

Reviewer 1 Report

The manuscript by Rome et al. presents the current knowledge on the role and potential clinical aplication of one of the subpopulations of extracellular vesicles, here, derived from adipocytes. The authors took up an interesting and valuable topic, undoubtedly worth further research. Some minor comments to the manuscript are listed below.

  1. After reading the manuscript, I have the impression that in many places there are minor repetitions and too general approach to the topic in question. sometimes the authors indicate the impact, but do not specify what it is, which leaves a slight dissatisfaction. The work done by the authors is impressive and worthy of recognition, nevertheless I suggest re-viewing the text in this aspect.
  2. the abbreviations used in the figures should be explained in the description below the figure
  3. the graphs of Fig. 3 are too small and therefore unreadable, difficult to interpret. Some axle signatures are cut off.
  4. Figures 4 and 5: the description of the figures should be enriched, in line with that contained in figure 1 or 2. The present descriptions are too general and do not sufficiently characterize the figure depicted.
  5. line 377: citations (28, 42, 43) should not be presented collectively as not all the factors listed below appear in each of the indicated works. In such a situation, it is more appropriate to add a column in the table and mark each reference factor with the appropriate reference.

Author Response

Response uploaded as a word file

Reviewer 2 Report

In this review paper authors have highlighted the role of adipocyte derived extracellular vesicles on disease progression and therapeutic targeting. There are many publication showing involvement of these vesicles in cell-to-cell communication. The topic is very intereting EVs is emerging research hot topic. The title of this review falls under the scope of IJMS journal. However, there are few issues in the current manuscript that authors need to address.

  1. Abstract first sentence “White adipose tissue (WAT) is the dedicated organ in our organism for long-term energy storage and represent 10-15% of total body weight in a healthy human”. Please re-write it. WAT is not an organ, but it found surrounding the organ and other place. It is just tissue and the phrase “organ in our organism” sounds odd.
  2. There are few plagiarised sentences. Please reduced % plagiarism
  3. Texts are cryptic in few places throughout the manuscript. Needs extensive grammatical corrections
  4. Introduction, first paragraph, line 29-38, please cite it.
  5. Introduction, second paragraph, line 39-41, please cite it.
  6. Line 47, define “low grade inflammation”
  7. Line 55-56 “We will also discuss about the AdEV components with may participate to AdEV biological effects” This sentence is not clear. “with may”?
  8. Line 56 “potential strategies” Please be specific which strategies? “therapeutic strategies”?
  9. Line 77, please elaborate more to understand “cytoskeleton remodeling”. Does this mean the role of flippase and floppases? Translocation of phosphotidylserine/choline from inner/outer membrane of cell? Check reference [4] and other relevant paper.
  10. Line 115: Define ESCRT
  11. Please elaborate more on clinical relevant difference in the content of AdEV released by adipocyte in healthy condition and disease state. Example: There might be different in protein or miRNA content in AdEV released by health and diabetic patient. (similar to normal and HFD mice model in figure 4).
  12. Section “Therapeutic strategies to decrease Ad EV deleterious effects” This section is well-elaborated. I was wondering EVs from two different source can show synergistic effect. For example: EVs from healthy WAT and EVs from mesenchymal stromal cells. Please discuss if any relevant study/citation.
  13. Although author have summarized section 4. “Therapeutic strategies to decrease Ad EV deleterious” in figure 5, but it looks too vague. I would recommend author to add table to including all relevant in vitro and in vivo study related to therapeutic strategies.

Author Response

Response uploaded as a word file

Reviewer 3 Report

A very detailed article on the role of EVs released by adipocytes. The manuscript is very well written.

I enclose the main comments that I ask the authors to take into account:

1.the article is written in the correct English, but it is worth having the language checked by a native speaker,
2. subchapter 2.1 would be presented as a separate chapter (this is just a small remark - if the authors feel that their version is better for the reader, I do not insist on changing it).
3. literature for the sentences lines 61-66 must be added: doi: 10.3389/fimmu.2018.02723.
4. the remaining chapters of the article are very well prepared and do not require correction.

Author Response

Response uploaded as a word file

Round 2

Reviewer 2 Report

Thank you for addressing most of my comments. The review looks better in quality and information. I have recommended for acceptance.

Author Response

Thanks